# POSITIONAL DESCRIPTION MATTERS FOR TRANSFORMERS ARITHMETIC

## ABSTRACT

Transformers, central to the successes in modern Natural Language Processing, often falter on arithmetic tasks despite their vast capabilities –which paradoxically include remarkable coding abilities. We observe that a crucial challenge is their naive reliance on positional information to solve arithmetic problems with a small number of digits, leading to poor performance on larger numbers. Herein, we delve deeper into the role of positional encoding, and propose several ways to fix the issue, either by modifying the positional encoding directly, or by modifying the representation of the arithmetic task to leverage standard positional encoding differently. We investigate the value of these modifications for three tasks: (i) classical multiplication, (ii) length extrapolation in addition, and (iii) addition in natural language context. For (i) we train a small model on a small dataset (100M parameters and 300k samples) with remarkable aptitude in (direct, no scratchpad) 15 digits multiplication and essentially perfect up to 12 digits, while usual training in this context would give a model failing at 4 digits multiplication. In the experiments on addition, we use a mere 120k samples to demonstrate: for (ii) extrapolation from 10 digits to testing on 12 digits numbers while usual training would have no extrapolation, and for (iii) almost perfect accuracy up to 5 digits while usual training would be correct only up to 3 digits (which is essentially memorization with a training set of 120k samples).

## 1 INTRODUCTION

In the world of Large Language Models (LLMs) advancements, arithmetic operations stand as a important yet frequently underestimated challenge. The emergence and triumph of models like GPT-4 (OpenAI, 2023) have had a transformative impact on various sectors, illuminating new potentials in Natural Language Processing and more. However, as we delve deeper into the diverse applications of these models, arithmetic tasks continually pose obstacles (Dziri et al., 2023), e.g., even GPT-4's struggles with tasks such as 4-digit multiplication.

Arithmetic tasks differ significantly from typical natural language tasks. The primary distinction lies in their execution: arithmetic operations might demand intricate intermediate steps internally, whereas most other tasks might not need such extensive internal processing. Furthermore, arithmetic operations possess a distinct data format. They utilize a concise vocabulary, but the potential combinations are vast. They also showcase more predictable patterns, and each token in an arithmetic sentence can hold equal significance. This contrasts with other tasks where omitting some non-essential words might not affect the overall meaning. Given the stark differences between arithmetic and other tasks, it's uncertain whether there's a straightforward way to bolster a language model's proficiency in arithmetic. Specifically, it's unclear if the prevailing architecture—tailored mainly for natural language tasks—can efficiently and accurately tackle arithmetic tasks. Moreover, this uniqueness of arithmetic also presents an opportunity: the structured nature of arithmetic, with its transparent steps and definitive outcomes, offers an ideal framework for a deeper understanding of the models. Addressing the challenges of arithmetic tasks and enhancing the arithmetic proficiency of LLMs can also contribute to a deeper understanding of their strengths and limitations.

In this work, we investigate the capabilities of language models concerning arithmetic operations, emphasizing techniques related to efficient position information utilization. Before delving into

our approaches, we first identify the important challenges that arithmetic tasks face. Three such challenges, central to this study, are:

**Complicated Calculation** Large-number multiplication and similar arithmetic tasks often involve intricate intermediate steps. Human solutions without using a calculator typically involve digit-wise multiplication, followed by summation. However, allowing a model to record each step can be verbose and inefficient for larger numbers. We investigate the feasibility of enabling a small transformer to directly output the product of large multiplication tasks.

**Length Extrapolation** Arithmetic data, unlike natural language data, typically exhibits highly regular patterns. As a result, models often depend on absolute position information to solve such tasks (Lee et al., 2023). For instance, in an addition operation like $a_1b_1c_1 + a_2b_2c_2$, aligning digits in corresponding positions (e.g., $a_1$ in position 1 and $a_2$ in position 5) presents the simplest solution. However, for a four-digit addition task like $a_1b_1c_1d_1 + a_2b_2c_2d_2$ that hasn't appeared in the training, it's unclear how to handle $d_1$ at position 4.

**Arithmetic and Language Integration** The poor performance of transformers on arithmetic data may be partly due to the lack of arithmetic data in the training set. However, it's uncertain whether simply supplementing the model with arithmetic data will resolve the problem. It's unclear if the model can successfully integrate arithmetic and natural language data due to their substantial differences.

We tackle the above challenges through two types of approaches, both aimed at utilizing position information more efficiently. A first approach is to alter the positional encoding directly. In this work, we explore an alternative positional encoding, namely randomized embedding, which is simple yet efficient for arithmetic tasks. A less direct approach for better position information utilization is to modify the representation of the arithmetic data to leverage standard positional encoding differently. We investigate how altering the data format can lead the model to learn the arithmetic task differently and exhibit varying properties.

In this work, we focus on small models of a GPT2-small size (124M). Our findings reveal that even such modest-sized models can adeptly execute intricate arithmetic tasks. This underscores not only the capability of the transformer architecture to handle arithmetic but also highlights that a small subset of model parameters can integrate arithmetic proficiency into language models, without affecting the model's capacity on other tasks.

We study large-number multiplication in Section 2, length generalization in Section 3 and arithmetic and language integration in Section 4. In this work, we tackle the three challenges outlined separately. However, in practice, we would need a single model that is able to show all the properties we want. This can be done by combining the approaches used in this paper, which we leave as a future work. For the purposes of this paper, we've maintained consistency in data size, model size, and training epochs, though it's conceivable that our observed outcomes could be achieved with reduced data sizes, smaller models, and fewer training epochs.

**Related Works**  Several recent works have studied using transformers to solve arithmetic tasks. Charton (2021; 2022) studied using transformers to do linear algebra. Zhang et al. (2022) studied variable assignment task. Qian et al. (2022) demonstrated the limitation of language models on arithmetic tasks. Hanna et al. (2023) studied the ability of GPT2 on arithmetic tasks from an interpretation viewpoint. Dziri et al. (2023) showed that even fine-tuned GPT3 has trouble performing 3-digit multiplication. Yang et al. (2023) trained a model of size 2B to perform arithmetic tasks and beat the performance of GPT4, but the accuracy obtained is not perfect even for 5-digit numbers. Lee et al. (2023) focused on the sample efficiency of using various data formats for arithmetic tasks and also studied the challenges we address in this paper, focusing on small numbers such as 3-digit addition and 2-digit multiplication. We are not aware of any previous work that is able to output the product of two 15-digit number multiplication, essentially perfect up to 12-digit, as demonstrated in our paper. Lee et al. (2023) also illustrates a model's ability to learn arithmetic and language simultaneously, but the two types of data remain separated.

A long list of works has focused on length generalization of transformers using a variety of positional encoding, including Su et al. (2021), Press et al. (2021), Li & McClelland (2022), Ruoss et al. (2023). Jelassi et al. (2023) shows that relative position embedding (Su et al., 2021), the encoder-only model can generalize to significantly longer lengths on arithmetic tasks.

To solve math problems using transformers, Uesato et al. (2022), Cobbe et al. (2021) and Lightman et al. (2023) used verifier and feedback. Zhou et al. (2022) used advanced prompting technique.

## 2 LARGE NUMBER MULTIPLICATION

Multiplication entails a sequence of intermediate steps, especially when dealing with large numbers. Modern language models like GPT-4 often find it challenging to handle these extensive multiplications (see Table 1). One test we can do is to ask the model to output the product directly, without using a scratchpad. We believe studying how the model can output the answer directly, bypassing intermediary steps, is an important research direction because in practice, outputting every step can be laborious and time-consuming. More importantly, always outputting the full steps can also prevent the model from using the most efficient method to solve the problem. In Section 2.1, we show a simple 12-layer transformer can output the product of $15 \times 15$-multiplication directly, demonstrating the immense potential of transformers. Constraining the model to use the scratchpad can force the model to adopt suboptimal strategies. While it can be hard for the model to learn to output the answers directly without using a scratchpad, our experiment indicates that given the right dataset and training regimen, it is feasible.

Large number multiplication is complicated, so it can be hard for the model to detect the rules for multiplication if we train the model directly with complicated multiplication tasks. However, there exist simple cases such as one-digit multiplications. By starting with these straightforward cases, the model can initially grasp rules from the basics and then extrapolate to more complex situations.

| # Digits | 1 | 2 | 3 | 4 | 5 |
|----------|------|------|------|------|------|
| 1 | 1.00 | 1.00 | 1.00 | 0.99 | 1.00 |
| 2 | 1.00 | 0.98 | 0.96 | 0.75 | 0.58 |
| 3 | 1.00 | 0.93 | 0.59 | 0.24 | 0.18 |
| 4 | 0.98 | 0.80 | 0.28 | 0.05 | 0.01 |
| 5 | 0.96 | 0.60 | 0.12 | 0.00 | 0.00 |

**Table 1:** Accuracy of GPT4 on 1-to-5-digit multiplcations. We use the prompt "What is $a * $b?", where $a and $b are the multiplicand and the multiplier. The row number represents the number of digits the multiplier has. The column number represents the number of digits the multiplicand has.

| # Digits | 1 | 2 | 3 | 4 | 5 |
|----------|------|------|------|------|------|
| 1 | 1.00 | 1.00 | 1.00 | 1.00 | 1.00 |
| 2 | 1.00 | 0.99 | 0.93 | 0.87 | 0.86 |
| 3 | 1.00 | 0.93 | 0.67 | 0.32 | 0.25 |
| 4 | 0.99 | 0.86 | 0.34 | 0.08 | 0.00 |
| 5 | 0.99 | 0.82 | 0.26 | 0.04 | 0.01 |

**Table 2:** Testing accuracy on models trained on 5-digit-maximum in basic format.

For our initial attempt, we included a lot of small-number multiplication in our datset. Our aim was to ensure the model had ample exposure to basic multiplications, enabling it to grasp multiplication rules. We create a dataset with 300k samples on 1-to-n-digit multiplication. We generate the two numbers in a way such that the number of digits of the two numbers is sampled uniformly randomly from $\{1, ..., n\}$. Although this uniform distribution ensures a balanced representation of numbers of different lengths, our emphasis leans towards smaller numbers. For example, our training set consists of around $8k$ single-digit number times a single-digit number, but there are only 100 different one-digit multiplications, so there will be a lot of repeated single-digit multiplication in our training set. On the contrary, the training set contains only less than $0.0002\%$ of 5-digit times 5-digit numbers. In the "Basic" format, the multiplier, multiplicant, and their product are presented straightforwardly. For instance, for two numbers 73866 and 1001, we write down "7 3 8 6 6 ∗ 1 0 0 1 # 7 3 9 3 9 8 6 6".
[1] We show in Table 2 the performance of a randomly initialized GPT2-small trained for 300 epochs when $n = 5$ and in Table 14 the performance when $n = 10$. The model performs well on 1-2-digit multiplication, but very poorly on large numbers. Notably, we see a trend that the model performs poorly when the sum of the number of digits in the two factors is greater than 5. When the sum is smaller than 5, the training set includes more than $10\%$ of all possible number combinations, leading to uncertainty regarding whether the model's proficiency with smaller numbers stems from genuine understanding or mere memorization.

Our findings show that emphasizing the small numbers is not enough for the model to perform well on large numbers. As the next step, we will focus on modifying the simple case, where the model

---

[1]In this paper, for every dataset used, a space is inserted before each digit. This ensures the tokenizer tokenizes each digit as an individual token.

can grasp the rule, so that the model can extrapolate to the hard cases efficiently. In Section 2.1 and Section A.1, we will present two distinct approaches designed to help the model draw connections between simpler and more complex tasks. These two approaches follow different principles and we hope they can inspire innovative simple case formulations not only for this multiplication task but for other tasks as well.

## 2.1 PADDING

For datapoints on multiplications of numbers with different numbers of digits, the position of the product sign varies. Consequently, the model needs to figure out the position of the product sign first and then perform the operation based on the relative position. This makes the rule of operation unnecessarily hard. A simple modification we can adopt is to add zero-padding to the training samples so that all the numbers are given in the same length. In this way, all multiplications will follow one rule no matter how many the number of digits the two factors have. If the maximum number of digits for the factors is $n$, we pad 0 so that both factors contain $n$ digits and the product contains $2n$ digit.

In addition, to make the task even easier, we can reverse the digits in the product. The rationale behind this is that to get the most significant digit of the product, we need to compute the carry from each digit accurately but to get the least significant digit, we only need to use the least significant digits of the two factors. As a result, starting with the least significant digit and progressing to the most significant digit is more straightforward. This intuitive approach has been used in previous works such as Lee et al. (2023).

| Data Format | Example |
|---|---|
| Basic | 7 3 8 6 6 * 1 0 0 1 # 7 3 9 3 9 8 6 6 |
| Reverse Product | 7 3 8 6 6 * 1 0 0 1 # 6 6 8 9 3 9 3 7 |
| Add Padding | 7 3 8 6 6 * 0 1 0 0 1 # 0 0 7 3 9 3 9 8 6 6 |
| Add Padding + Reverse Product | 7 3 8 6 6 * 0 1 0 0 1 # 6 6 8 9 3 9 3 7 0 0 |

**Table 3:** Examples of the data format for multiplication.

**(a)** Add Padding

| # Digits | 1 | 2 | 3 | 4 | 5 |
|---|---|---|---|---|---|
| 1 | 1.00 | 1.00 | 1.00 | 1.00 | 0.99 |
| 2 | 1.00 | 0.99 | 0.99 | 0.94 | 0.88 |
| 3 | 1.00 | 0.99 | 0.86 | 0.82 | 0.75 |
| 4 | 0.99 | 0.96 | 0.79 | 0.73 | 0.68 |
| 5 | 1.00 | 0.90 | 0.72 | 0.62 | 0.59 |

**(b)** Reverse Product

| # Digits | 1 | 2 | 3 | 4 | 5 |
|---|---|---|---|---|---|
| 1 | 1.00 | 1.00 | 1.00 | 1.00 | 1.00 |
| 2 | 1.00 | 1.00 | 1.00 | 1.00 | 1.00 |
| 3 | 1.00 | 1.00 | 1.00 | 0.97 | 0.28 |
| 4 | 1.00 | 1.00 | 0.97 | 0.28 | 0.02 |
| 5 | 1.00 | 0.99 | 0.29 | 0.06 | 0.02 |

**(c)** Reverse Product + Add Padding

| # Digits | 1 | 2 | 3 | 4 | 5 |
|---|---|---|---|---|---|
| 1 | 1.00 | 1.00 | 1.00 | 1.00 | 1.00 |
| 2 | 1.00 | 1.00 | 1.00 | 1.00 | 1.00 |
| 3 | 1.00 | 1.00 | 1.00 | 1.00 | 1.00 |
| 4 | 1.00 | 1.00 | 1.00 | 1.00 | 1.00 |
| 5 | 1.00 | 1.00 | 1.00 | 1.00 | 1.00 |

**Table 4:** Testing accuracy for models trained on data with padding and(or) reversed product when the maximum number of digits is 5.

| # Digits | 1 | 2 | 3 | 4 | 5 | 6 | 7 | 8 | 9 | 10 | 11 | 12 | 13 | 14 | 15 |
|---|---|---|---|---|---|---|---|---|---|---|---|---|---|---|---|
| 1 | 1.00 | 1.00 | 1.00 | 1.00 | 1.00 | 1.00 | 1.00 | 1.00 | 1.00 | 1.00 | 1.00 | 1.00 | 1.00 | 1.00 | 1.00 |
| 2 | 1.00 | 1.00 | 1.00 | 1.00 | 1.00 | 1.00 | 1.00 | 1.00 | 1.00 | 1.00 | 1.00 | 1.00 | 1.00 | 1.00 | 1.00 |
| 3 | 1.00 | 1.00 | 1.00 | 1.00 | 1.00 | 1.00 | 1.00 | 1.00 | 1.00 | 1.00 | 1.00 | 1.00 | 1.00 | 1.00 | 1.00 |
| 4 | 1.00 | 1.00 | 1.00 | 1.00 | 1.00 | 1.00 | 1.00 | 1.00 | 1.00 | 1.00 | 1.00 | 1.00 | 1.00 | 1.00 | 1.00 |
| 5 | 1.00 | 1.00 | 1.00 | 1.00 | 1.00 | 1.00 | 1.00 | 1.00 | 1.00 | 1.00 | 1.00 | 1.00 | 1.00 | 1.00 | 1.00 |
| 6 | 1.00 | 1.00 | 1.00 | 1.00 | 1.00 | 1.00 | 1.00 | 1.00 | 1.00 | 1.00 | 1.00 | 1.00 | 1.00 | 1.00 | 1.00 |
| 7 | 1.00 | 1.00 | 1.00 | 1.00 | 1.00 | 1.00 | 1.00 | 1.00 | 1.00 | 1.00 | 1.00 | 1.00 | 1.00 | 1.00 | 1.00 |
| 8 | 1.00 | 1.00 | 1.00 | 1.00 | 1.00 | 1.00 | 1.00 | 1.00 | 1.00 | 1.00 | 1.00 | 1.00 | 1.00 | 1.00 | 1.00 |
| 9 | 1.00 | 1.00 | 1.00 | 1.00 | 1.00 | 1.00 | 1.00 | 1.00 | 1.00 | 1.00 | 1.00 | 1.00 | 1.00 | 1.00 | 1.00 |
| 10 | 1.00 | 1.00 | 1.00 | 1.00 | 1.00 | 1.00 | 1.00 | 1.00 | 0.99 | 1.00 | 1.00 | 1.00 | 1.00 | 1.00 | 0.99 |
| 11 | 1.00 | 1.00 | 1.00 | 1.00 | 1.00 | 1.00 | 1.00 | 1.00 | 1.00 | 1.00 | 1.00 | 1.00 | 1.00 | 1.00 | 0.99 |
| 12 | 1.00 | 1.00 | 1.00 | 1.00 | 1.00 | 1.00 | 1.00 | 1.00 | 0.99 | 1.00 | 1.00 | 1.00 | 1.00 | 0.99 | 0.99 |
| 13 | 1.00 | 1.00 | 1.00 | 1.00 | 1.00 | 1.00 | 1.00 | 1.00 | 1.00 | 1.00 | 0.99 | 0.99 | 0.95 | 0.95 | 0.95 |
| 14 | 1.00 | 1.00 | 1.00 | 1.00 | 1.00 | 1.00 | 1.00 | 1.00 | 1.00 | 1.00 | 0.99 | 1.00 | 0.98 | 0.95 | 0.93 |
| 15 | 1.00 | 1.00 | 1.00 | 1.00 | 1.00 | 1.00 | 1.00 | 1.00 | 1.00 | 1.00 | 0.96 | 0.98 | 0.95 | 0.98 | 0.84 |

**Table 5:** Testing accuracy for 15-maximum-digit with padding and reversed product.

In Table 3, we present examples of our data format. We give more details on the dataset and the setup in Appendix B. The accuracy by GPT2-small on 300k samples achieved using padding and/or reversed product for multiplications with a maximum of 5 and 10 digits is detailed in Table 4 and Table 15 respectively. The results indicate that padding markedly boosts accuracy while reversing

the product further elevates it to near perfection. Utilizing both padding and reversed product allows us to accurately compute up to $15 \times 15$ multiplications, as shown in Table 5. This is a remarkable enhancement when compared to the non-padded data format, which encountered difficulties even with $4 \times 4$ multiplication. The benefit of padding is that it standardizes the format between basic and more complex cases, enabling them to be addressed by a singular rule and enhancing the link between them.

However, when evaluating accuracy for a maximum of 20 digits in Table 16, the results for larger numbers are unsatisfactory. We did not fine-tune the parameters in our experiment, so it is possible we can achieve high accuracy for even more digits if we use a larger training set, a more optimal digit distribution, or a more fine-tuned learning rate, etc.

## 3 LENGTH EXTRAPOLATION

In this section, we tackle a different challenge from the previous section, length extrapolation. While relying on position information can help complicated arithmetic tasks, overreliance on position can hurt generalization to additional digits. Based on the idea of reducing the reliance on absolute positional information, in Section 3.1, we delve into various data formats that can help generalization to additional digits, and in section 3.2, we investigate the role of vanilla positional embedding in arithmetic tasks and explore alternative positional embedding that better suits the needs of arithmetic tasks.

### 3.1 DATA FORMAT

In this section, we explore the impact of different data formats on the generalization performance of models when faced with additional digits in the addition task. We propose two distinct data formats that aid in improving the models' ability to generalize. One straightforward data format is a chain-of-thought (Wei et al., 2022) style scratchpad. In this format, we first write down the two addends of the addition, followed by the digit-wise summation steps and the final sum. However, as expected, this format struggles to generalize to numbers longer than those encountered during training. A common mistake made by the model is omitting digits while recording the digit-wise summation steps. To address this issue, we explore new data formats based on two key ideas.

The first idea involves introducing random spaces between characters in the data. By doing so, we make it more challenging for the model to rely on absolute positional embedding to solve the task. This disruption encourages the model to consider other cues beyond the absolute positional information.

The second idea is based on repeating more information for each digit-wise step. This approach allows the model to access additional information, enabling it to learn the actual steps rather than solely relying on memorizing positional patterns. The increased redundancy makes it harder for the model to overfit to positional information. We found that data formats based on both of these ideas significantly improve generalization performance. By incorporating random spaces and increasing information repetition, the models gain the ability to better handle numbers with more digits and exhibit enhanced generalization performance.

We test our two ideas on three data formats. Table 7 shows the examples of these three data formats, where random space is based on the first idea and recursive scratchpad is based on the second idea. We give the formal definition of the data formats and the setup in Appendix C.1. We show in Table 6 the accuracy of the model on the three types of data formats. We further give a few failure examples of the models trained on each data format in Table 17. Our experimental results corroborate our conjectures.

In the "Basic" data format, the model fails to generalize numbers exceeding 10 digits. If the model is given two addends that exceed this length, it simply omits some digits and outputs a result with only 10 steps. However, incorporating random spaces into the training set compels the model to move away from relying on absolute positional embedding since it can no longer retrieve the digits from fixed positions. Despite the model's accurate prediction only extending by one digit, this progression represents a significant improvement, demonstrating a phase transition from a complete lack of generalization to some degree of it. We observe an even more significant improvement in

generalization performance when we increase the information provided in each digit-wise step. This suggests that adding more information can encourage the model to learn the fundamental recursive steps required for addition, as opposed to overfitting to positional information.

| Data Format | 9 Digits | 10 Digits | 11 Digits | 12 Digits | 13 Digits |
|---|---|---|---|---|---|
| Basic | 1.00 | 1.00 | 0.00 | 0.00 | 0.00 |
| Random Space | 1.00 | 1.00 | 0.99 | 0.09 | 0.00 |
| Recursive Scratchpad | 1.00 | 1.00 | 1.00 | 0.96 | 0.55 |

**Table 6:** Testing accuracies on 9-13-digit-addition of models trained on the three data formats of 2-10-digit-addition.

| Data Format | Example |
|---|---|
| Basic | 2 3 9 + 8 2 1 : 0 + 9 + 1 = 1 0, 1 + 3 + 2 = 6, 0 + 2 + 8 = 1 0, 1 0 6 0 |
| Random Space | 2   3 9  + 8  2 1 :  0 +  9 + 1 = 1  0, 1 +  3 + 2  = 6,  0 + 2 +  8 = 1 0, 1 0  6 0 |
| Recursive Scratchpad | 2 3 9 + 8 2 1 : 0 + 9 + 1 = 1 0 , = 0 , 3 2 + 2 8 : 1 + 3 + 2 = 6 , = 6 0 , 2 + 8 : 0 + 2 + 8 = 1 0 , = 0 6 0 , = 1 0 6 0 |

**Table 7:** Examples of the data format for adding 239 and 821.

In addition, we would like to make the following remarks.

**Pretrained vs. Randomly Initialized** We found that in this task, using a pretrained model is important for "recursive scratchpad". Without using a pretrained model, "recursive scratchpad" won't help generalization to additional digits. However, it does not make much difference for "random space". For both pretrained and randomly initialized models, "basic" does not generalize to additional digits. We will have more discussion on training from scratch on the addition task in Section 3.2.

**Reverse the order of the digits in the addends** For "Recursive Scratchpad", we found that reversing the order of the digits of the addends can help the generalization performance. However, reversing the order of both the addends and the sum will not help as much.

## 3.2 POSITIONAL EMBEDDING

As we discussed in Section 3.1, the data format can greatly influence a model's dependency on positional information, which subsequently affects its generalization capacity. In this section, we directly examine positional embedding by studying its limitations and exploring potential alternatives.

To better understand the significance of positional embedding, we first consider a simpler task: given a number, the model outputs its digits in reverse order. For instance, if the input number is $12345$, the output should be $54321$. We evaluate the model's performance on numbers that are longer than those in the training set and investigate challenging cases such as numbers with many repeated digits. We give the formal definition of the two types of data in Appendix C.2.

As an initial step, we eliminated the positional embedding of the GPT2-small while leaving the rest of the architecture intact. It appears that for both the pre-trained model and the model trained from scratch, the removal of positional embedding enhances the generalization capacity across more digits. We show in Figure 1 the test accuracy of both models on regular and repetitive data. Figure 1a indicates that upon deletion of the positional embedding, both models exhibit an improvement in generalization by approximately two digits on the regular data. While we don't observe a significant accuracy discrepancy between the two models on regular data, their performance on repetitive data varies considerably. As shown in Figure 1b, the repetitive data does not pose a difficult challenge for the model with positional embedding. However, it becomes notably difficult for the model trained from scratch, which achieves low accuracy even with 9-digit data. In contrast, it's relatively simple for the pre-trained model, which manages to achieve perfect accuracy with 16-digit data. We speculate that the underlying reason is the inability to differentiate repetitive data aside from their respective positions. Without absolute positional embedding, the models must resort to alternative methods to encode positional information. Given that the pre-trained model already contains various useful pre-trained components, it has greater flexibility to address this issue.

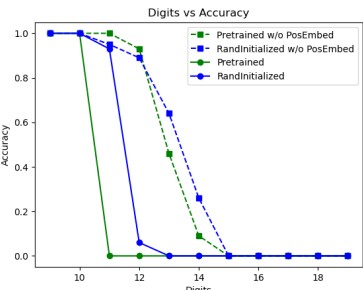 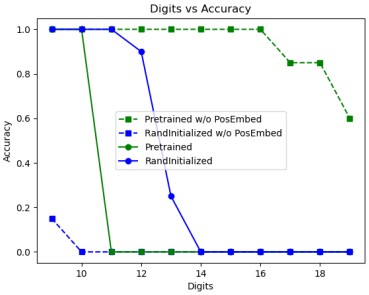

**(a)** Test accuracy on regular data.    **(b)** Test accuracy on repetitive data.

**Figure 1:** Comparison of pretrained model and trained from the scratch model with and without absolute positional embedding on 100 regular testing samples and repetitive samples. We use pretrained and random initialized GPT2-small with/without the positional embedding and fine-tune/train for 10 epochs with a learning rate 2e-5.

To solve this issue, we propose a simple solution targeting this issue. We mark each token using a random tag so that the model can easily use the tag to distinguish the same tokens appearing at different positions. We call this component a random embedding. We are able to show that this random tag can not only improve the generalization performance on this simple task of digit reversion but also on the more complicated task of addition.

**Random Embedding**  For any chosen hash dimension $n_{\text{hash}}$, we generate a $n_{\text{hash}}$-dimensional random Gaussian vector with mean 0 and identity covariance. Then, we split the Gaussian vector into $n_{\text{head}}$ many vectors $\{h_i\}_{i=1}^{n_{\text{head}}}$ each with dimension $n_{\text{hash}}/n_{\text{head}}$, set the last $n_{\text{hash}}/n_{\text{head}}$ dimensions of the input embedding of each head to be $h_i$ and keep the remaining $(n_{\text{embed}} - n_{\text{hash}})/n_{\text{head}}$ dimensions unchanged. After the final layer, we use only the first $(n_{\text{embed}} - n_{\text{hash}})/n_{\text{head}}$ dimension of each head to decode. We use newly generated random vectors for each epoch and during testing.

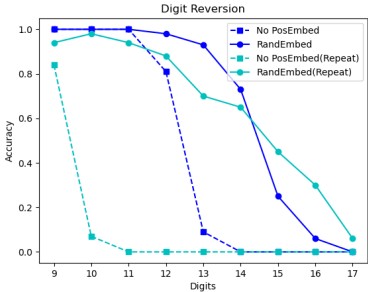 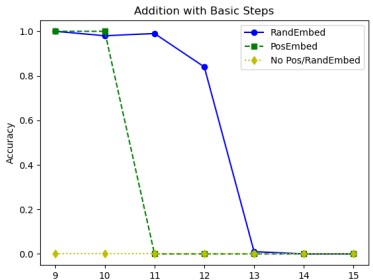

**(a)** Test accuracy on repetitive data in the digit reversion task for models trained from randomly initialized weights.

**(b)** Test accuracy on addition with "Basic" step task for models trained from randomly initialized weights.

**Figure 2:** Comparison of trained from scratch model with and without hash embedding on 100 regular testing samples and repetitive samples. We use random initialized GPT2-small (124M) without the positional embedding and train for 25 epochs with a learning rate 1e-5.[2]

In Figure 2, we demonstrate the improved generalization capacity of GPT2-small equipped with random embedding. Figure 2a shows that adding random embedding increases the generalization capacity on both the regular data and the repetitive data in the digit reversion task.

Back to the more complicated task of addition, we show in Figure 2b that if we simply delete the positional embedding, the random initialized model does not perform well. If we keep the positional

---

[2]The accuracies of "No PosEmbed" differ slightly from the corresponding accuracies in Figure 1 because the accuracies are measured from different runs with different learning rates and numbers of epochs.

embedding, the model does not generalize to more digits. The random embedding shows significant improvement by achieving about the same generalization capacity as the "Recursive Scratchpad" data format as we show in Section 3.1.

## 4  ADDITION IN NATURAL LANGUAGE SETTING

In the previous sections, we focused on the case where the training data consists of solely arithmetic data. However, in practice, we need to do the arithmetic operations in the natural language setting. Training data consisting exclusively of arithmetic data is usually easy to collect as it can be generated programmatically in large quantities. On the contrary, obtaining arithmetic information embedded in natural language is a more arduous task due to its rarity in natural language content. Consequently, it is important to understand if training on purely arithmetic data can equip the model with the ability to perform arithmetic tasks within a natural language setting.

In this section, we explore a task that involves mixing natural language data with purely arithmetic data to investigate the model's ability to integrate both data types. The natural language data in this case includes dialogues on solving addition problems, with a substantial amount of samples for easy addition questions and a smaller portion for difficult ones. Such dataset structure reflects the real-world challenge of readily collecting easy tasks, while struggling to find natural language data that solves more complex problems. Alongside this, we incorporate purely arithmetic data, which is always correct and can be effortlessly produced using computer programs. Our primary objective is to examine whether the accurate arithmetic data can help the model solve the complex tasks embedded in the natural language context.

Our experiments show that in our setting, training solely on arithmetic data can't guarantee an accurate solution to the difficult problems due to the lack of difficult samples in the training set and the errors present. If we naively mix the arithmetic data with the natural language data, we don't see a significant boost in accuracy, which shows that integrating the two types of data is challenging if they follow different formats. One obvious difficulty arises when the arithmetic data follows a certain pattern; the model can easily learn the arithmetic task relying on the positional information. However, when the arithmetic task appears in the natural language context, it won't follow the same positional pattern, causing the model to struggle to connect the two types of data. Overreliance on positional embedding is a recurring issue when using transformers for arithmetic tasks, and this represents the main challenge we discuss in Section 3. In Section 3, we tackle this issue from two aspects: data format and alternative position embedding. We show in our experiments that similar ideas can be applied to the integration of natural language and arithmetic data, thus facilitating the merging of these two types of data.

| Data Format | Examples |
|---|---|
| Dialogue Data (Dia) | Student: Excuse me, can you help me with something? I need to add two numbers, 842 and 62. Teacher: Of course, let me do the calculation for you. The answer is 904. |
| | Student: Good morning! Can you help me with a math problem? I need to find the sum of 324 and 402. Teacher: Good morning! Sure thing. The answer is 726. |
| Addition Data - Basic | 4 8 + 4 = 5 2 |
| | 3 7 5 + 2 6 1 = 6 3 6 |
| | 5 0 5 1 + 8 5 3 9 = 1 3 5 9 0 |
| Addition Data - Random Space | 4 8    4    5 2 |
| | 3 7 5    2 6 1    6 3 6 |
| | 5 0 5 1    8 5 3 9    1 3 5 9 0 |

**Table 8:** Examples of the natural language and arithmetic data used.

We use three types of data formats, formally defined in Appendix D with examples shown in Table 9. Our dialogue dataset contains a large number of 2-3-digit addition, but not enough 4-5-digit addition while the addition data set contain a large number of both 2-3-digit addition and 4-5-digit addition. We compare in Table 9 models trained on datasets that combine dialogue data with addition data (Dia+Basic and Dia+RandomSpace) to those trained solely on dialogue data (Dia).

| #Digit | NL | | | | NL+Basic | | | | NL+Random Space | | | |
|---|---|---|---|---|---|---|---|---|---|---|---|---|
| | 2 | 3 | 4 | 5 | 2 | 3 | 4 | 5 | 2 | 3 | 4 | 5 |
| PosEmbed | 0.94 | 0.92 | 0.61 | 0.13 | 1.0 | 0.96 | 0.68 | 0.12 | 1.0 | 0.99 | 0.91 | 0.82 |
| NoPosEmbed | 0.9 | 0.97 | 0.67 | 0.11 | 0.99 | 1.0 | 0.99 | 0.99 | 0.96 | 0.99 | 1.0 | 0.99 |
| RandEmbed | 0.9 | 0.85 | 0.57 | 0.07 | 1.0 | 1.0 | 0.97 | 0.86 | 1.0 | 1.0 | 0.96 | 0.91 |

**(a)** Testing accuracy in the dialogue context for models trained for 100 epochs.

| #Digit | NL | | | | NL+Basic | | | | NL+Random Space | | | |
|---|---|---|---|---|---|---|---|---|---|---|---|---|
| | 2 | 3 | 4 | 5 | 2 | 3 | 4 | 5 | 2 | 3 | 4 | 5 |
| PosEmbed | 0.97 | 0.96 | 0.62 | 0.14 | 0.99 | 0.94 | 0.64 | 0.07 | 0.99 | 0.98 | 0.93 | 0.77 |
| NoPosEmbed | 0.83 | 0.96 | 0.47 | 0.08 | 1.0 | 1.0 | 0.98 | 0.6 | 1.0 | 1.0 | 0.99 | 0.76 |
| RandEmbed | 0.77 | 0.69 | 0.13 | 0.0 | 1.0 | 0.98 | 0.91 | 0.36 | 0.99 | 0.99 | 0.89 | 0.83 |

**(b)** Testing accuracy in the dialogue context for models trained for 50 epochs.

| #Digit | NL+Basic | | | | NL+Random Space | | | |
|---|---|---|---|---|---|---|---|---|
| | 2 | 3 | 4 | 5 | 2 | 3 | 4 | 5 |
| PosEmbed | 0.99 | 1.0 | 0.98 | 0.98 | 1.0 | 1.0 | 1.0 | 1.0 |
| NoPosEmbed | 0.2 | 0.08 | 0.04 | 0.01 | 0.95 | 0.96 | 0.96 | 0.27 |
| RandEmbed | 0.99 | 1.0 | 1.0 | 1.0 | 1.0 | 1.0 | 1.0 | 1.0 |

**(c)** Testing accuracy in the pure addition context for models trained for 100 epochs.

| #Digit | NL+Basic | | | | NL+Random Space | | | |
|---|---|---|---|---|---|---|---|---|
| | 2 | 3 | 4 | 5 | 2 | 3 | 4 | 5 |
| PosEmbed | 0.99 | 1.0 | 1.0 | 0.99 | 1.0 | 1.0 | 1.0 | 1.0 |
| NoPosEmbed | 0.91 | 0.77 | 0.4 | 0.22 | 0.98 | 1.0 | 0.98 | 0.78 |
| RandEmbed | 0.99 | 1.0 | 1.0 | 0.98 | 0.98 | 1.0 | 0.99 | 0.97 |

**(d)** Testing accuracy in the pure addition context for models trained for 50 epochs

**Table 9:** Testing accuracy for models with and without the positional embedding and with the random embedding on the dataset solely consists of dialogue data and the data set consists of a mix of dialogue data and addition data.

We show the results for random initialized models trained 50 epochs and 100 epochs. Without any arithmetic data, models trained exclusively on dialogue struggle to accurately perform 4-5-digit addition. This confirms our hypothesis, given that the dialogue lacks a sufficient number of correct 4-5-digit examples. With arithmetic data, for models with the absolute position embedding, "Basic" doesn't significantly enhance their ability to tackle addition tasks within the dialogue prompt. In contrast, using "Random space", removing the absolute position embedding, and integrating random embedding all improve the model's ability to leverage addition data in supporting dialogue-based addition tasks. For models that exclude absolute position embedding, as well as those with random embedding, the testing accuracy for "Basic" and "Random space" is similar when trained for long enough. Nevertheless, models can learn the "Random space" format slightly faster, as shown in Table 9a and Table 9b. Models without position embedding exhibit slightly better accuracy compared to those with random embedding in dialogue contexts. Conversely, models with random embedding outperform those lacking position embedding in pure addition scenarios, as highlighted in Table 9c and Table 9d.

In conclusion, to allow language and arithmetic integration, we need either data format modification such as random space, or position embedding modification such as excluding absolute positional embedding or adding random embedding. Our conclusions here align with those in Section 3. For models with absolute position embedding, the "Basic" format is less effective due to its highly predictable pattern, allowing models to overly depend on positional information. Removing position embedding addresses this, but can create new stability issues as the model needs alternative ways to interpret position data. Introducing random embedding can offset the drawbacks of removing position embedding, resulting in a more stable performance.

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

# A  ADDITIONAL DETAILS ON LARGE NUMBER MULTIPLICATION IN SECTION 2

## A.1  $n \times 1$-MULTIPLICATION

**(a)** $n \times 1$-multiplication, 5-maximum-digit

| # Digits | 1 | 2 | 3 | 4 | 5 |
|---|---|---|---|---|---|
| 1 | 1.00 | 1.00 | 1.00 | 1.00 | 1.00 |
| 2 | 1.00 | 1.00 | 1.00 | 1.00 | 1.00 |
| 3 | 1.00 | 1.00 | 1.00 | 1.00 | 1.00 |
| 4 | 1.00 | 1.00 | 1.00 | 0.84 | 0.59 |
| 5 | 1.00 | 1.00 | 1.00 | 0.58 | 0.17 |

**(b)** $n \times 1$-multiplication, 6-maximum-digit

| # Digits | 1 | 2 | 3 | 4 | 5 | 6 |
|---|---|---|---|---|---|---|
| 1 | 1.00 | 1.00 | 1.00 | 1.00 | 1.00 | 1.00 |
| 2 | 1.00 | 1.00 | 1.00 | 1.00 | 1.00 | 1.00 |
| 3 | 1.00 | 1.00 | 0.86 | 0.58 | 0.49 | 0.15 |
| 4 | 1.00 | 1.00 | 0.70 | 0.13 | 0.05 | 0.01 |
| 5 | 1.00 | 1.00 | 0.53 | 0.05 | 0.01 | 0.00 |
| 6 | 1.00 | 1.00 | 0.26 | 0.03 | 0.01 | 0.00 |

**(c)** First-step multiplication, 5-maximum-digit

| # Digits | 1 | 2 | 3 | 4 | 5 |
|---|---|---|---|---|---|
| 1 | 1.00 | 1.00 | 1.00 | 1.00 | 1.00 |
| 2 | 1.00 | 1.00 | 1.00 | 1.00 | 1.00 |
| 3 | 1.00 | 1.00 | 1.00 | 1.00 | 1.00 |
| 4 | 1.00 | 1.00 | 1.00 | 1.00 | 1.00 |
| 5 | 1.00 | 1.00 | 1.00 | 1.00 | 0.98 |

**(d)** First-step multiplication, 6-maximum-digit

| # Digits | 1 | 2 | 3 | 4 | 5 | 6 |
|---|---|---|---|---|---|---|
| 1 | 1.00 | 1.00 | 1.00 | 1.00 | 1.00 | 1.00 |
| 2 | 1.00 | 1.00 | 1.00 | 1.00 | 1.00 | 1.00 |
| 3 | 1.00 | 1.00 | 0.98 | 0.90 | 0.74 | 0.39 |
| 4 | 1.00 | 1.00 | 0.88 | 0.24 | 0.10 | 0.03 |
| 5 | 1.00 | 1.00 | 0.76 | 0.09 | 0.01 | 0.01 |
| 6 | 1.00 | 1.00 | 0.60 | 0.04 | 0.01 | 0.00 |

**Table 10:** Testing accuracies for models trained on dataset with $n \times 1$-multiplication every 3 datapoints and first-step multiplication every 3 datapoints.

In multiplication, $n \times 1$-multiplication serves as a very important simple case. If our dataset does not contain enough $n \times 1$-multiplication, the models will not be able to learn multiplication successfully even with padding and reversed product. We show in Figure 11 how decreasing the ratio of $n \times 1$-multiplication hurts the model's performance. With this insight, we amplify the presence of $n \times 1$ multiplication in the training set, hypothesizing that this could potentially improve multiplication accuracy, even in the absence of padding. As an initial step, for every three data points of the dataset, we modify the second factor of the multiplication to a single-digit number. In other words, given the original dataset, we choose $1/3$ of the datapoints $a * b$ and replace $b$ with $b\%10$. In this way, our dataset consists of a large number of $n \times 1$ multiplication. We give an example of such a dataset with n × 1 multiplication every 3 datapoints in Table 12. Our training datapoints have reversed product but no padding. We show the accuracy on 5-maximum-digit and 6-maximum-digit in Table 10a and 10b. A comparison between Table 10a and Table 4 reveals that this augmentation does somewhat enhance the model accuracy.

To use the simple $n \times 1$-multiplication data, the model must recognize that $n \times m$-multiplication can be decomposed into $m$ $n \times 1$ multiplication and use the knowledge gained from $n \times 1$-multiplication to solve the problem. To better bridge the understanding between $n \times 1$ and $n \times m$ multiplications, we undertook a more bold adjustment to our dataset. Rather than merely incorporating more $n \times 1$ multiplications, we introduced an entirely new type of data that explicitly tells the model that $n \times 1$ multiplication is a foundational step for $n \times m$ multiplication. To this end, we introduce a new operation $\%$ where for two numbers $a$ and $b$, we define $a\%b = a \times (b \mod 10)$. In this way, we can view $\%$ as guidance on the first step of solving $a * b$. We change the operation from $*$ to $\%$ for every 3 datapoints in our training set. We show in Table 12 examples of data with first-step multiplication $\%$ every 3 datapoints. We show in Table 10c and 10d the accuracy on 5-maximum-digit and 6-maximum-digit. We see a further performance improvement from Table 10a and Table 10b.

While the enhancements observed from introducing the $\%$ operation might not be as pronounced as those achieved with padding, as discussed in Section 2.1, the outcomes of this experiment are both intriguing and enlightening. When comparing the two experiments – one adding additional $n \times 1$ multiplications, and the other integrating first-step multiplications – the latter introduces a more intricate dataset. This dataset comprises two markedly distinct types of data, requiring the models to discern the function of the novel $\%$ operation and subsequently correlate this operation to the first

**(a)** 1-digit weight = 0.4

| # Digits | 1 | 2 | 3 | 4 | 5 | 6 | 7 | 8 | 9 | 10 |
|---|---|---|---|---|---|---|---|---|---|---|
| 1 | 1.00 | 1.00 | 1.00 | 1.00 | 1.00 | 1.00 | 1.00 | 1.00 | 1.00 | 1.00 |
| 2 | 1.00 | 1.00 | 1.00 | 1.00 | 1.00 | 1.00 | 1.00 | 1.00 | 1.00 | 1.00 |
| 3 | 1.00 | 1.00 | 1.00 | 1.00 | 1.00 | 1.00 | 1.00 | 1.00 | 1.00 | 1.00 |
| 4 | 1.00 | 1.00 | 1.00 | 1.00 | 1.00 | 1.00 | 1.00 | 1.00 | 1.00 | 1.00 |
| 5 | 1.00 | 1.00 | 1.00 | 1.00 | 1.00 | 1.00 | 1.00 | 1.00 | 1.00 | 1.00 |
| 6 | 1.00 | 1.00 | 1.00 | 1.00 | 1.00 | 1.00 | 1.00 | 1.00 | 1.00 | 1.00 |
| 7 | 1.00 | 1.00 | 1.00 | 1.00 | 1.00 | 1.00 | 1.00 | 1.00 | 1.00 | 1.00 |
| 8 | 1.00 | 1.00 | 1.00 | 1.00 | 1.00 | 1.00 | 0.99 | 1.00 | 0.99 | 1.00 |
| 9 | 1.00 | 1.00 | 1.00 | 1.00 | 1.00 | 1.00 | 1.00 | 0.97 | 0.99 | 0.96 |
| 10 | 1.00 | 1.00 | 1.00 | 1.00 | 1.00 | 1.00 | 1.00 | 0.98 | 0.93 | 0.97 |

**(b)** 1-digit weight = 0.2

| # Digits | 1 | 2 | 3 | 4 | 5 | 6 | 7 | 8 | 9 | 10 |
|---|---|---|---|---|---|---|---|---|---|---|
| 1 | 1.00 | 1.00 | 1.00 | 1.00 | 1.00 | 1.00 | 1.00 | 1.00 | 1.00 | 1.00 |
| 2 | 1.00 | 1.00 | 1.00 | 1.00 | 1.00 | 1.00 | 1.00 | 1.00 | 1.00 | 0.93 |
| 3 | 1.00 | 1.00 | 1.00 | 1.00 | 1.00 | 1.00 | 1.00 | 0.98 | 0.94 | 0.13 |
| 4 | 1.00 | 1.00 | 1.00 | 1.00 | 1.00 | 0.98 | 0.33 | 0.06 | 0.02 | 0.00 |
| 5 | 1.00 | 1.00 | 1.00 | 1.00 | 0.97 | 0.26 | 0.04 | 0.01 | 0.00 | 0.00 |
| 6 | 1.00 | 1.00 | 1.00 | 0.97 | 0.21 | 0.01 | 0.01 | 0.00 | 0.00 | 0.00 |
| 7 | 1.00 | 1.00 | 1.00 | 0.24 | 0.07 | 0.00 | 0.00 | 0.00 | 0.00 | 0.00 |
| 8 | 1.00 | 1.00 | 0.99 | 0.11 | 0.00 | 0.00 | 0.00 | 0.00 | 0.00 | 0.00 |
| 9 | 1.00 | 1.00 | 0.98 | 0.01 | 0.00 | 0.00 | 0.00 | 0.00 | 0.00 | 0.00 |
| 10 | 1.00 | 0.93 | 0.09 | 0.00 | 0.00 | 0.00 | 0.00 | 0.00 | 0.00 | 0.00 |

**(c)** 1-digit weight = 0.0

| # Digits | 1 | 2 | 3 | 4 | 5 | 6 | 7 | 8 | 9 | 10 |
|---|---|---|---|---|---|---|---|---|---|---|
| 1 | 0.28 | 0.10 | 0.03 | 0.01 | 0.00 | 0.00 | 0.00 | 0.00 | 0.00 | 0.00 |
| 2 | 0.10 | 0.63 | 0.46 | 0.23 | 0.09 | 0.01 | 0.00 | 0.00 | 0.00 | 0.00 |
| 3 | 0.04 | 0.51 | 0.18 | 0.13 | 0.01 | 0.00 | 0.00 | 0.00 | 0.00 | 0.00 |
| 4 | 0.00 | 0.21 | 0.04 | 0.00 | 0.00 | 0.00 | 0.00 | 0.00 | 0.00 | 0.00 |
| 5 | 0.00 | 0.07 | 0.01 | 0.00 | 0.00 | 0.00 | 0.00 | 0.00 | 0.00 | 0.00 |
| 6 | 0.00 | 0.00 | 0.00 | 0.00 | 0.00 | 0.00 | 0.00 | 0.00 | 0.00 | 0.00 |
| 7 | 0.00 | 0.00 | 0.00 | 0.00 | 0.00 | 0.00 | 0.00 | 0.00 | 0.00 | 0.00 |
| 8 | 0.00 | 0.00 | 0.00 | 0.00 | 0.00 | 0.00 | 0.00 | 0.00 | 0.00 | 0.00 |
| 9 | 0.00 | 0.00 | 0.00 | 0.00 | 0.00 | 0.00 | 0.00 | 0.00 | 0.00 | 0.00 |
| 10 | 0.00 | 0.00 | 0.00 | 0.00 | 0.00 | 0.00 | 0.00 | 0.00 | 0.00 | 0.00 |

**Table 11:** Testing accuracy for 10-maximum-digit with padding and reversed product and varying 1-digit weight $\alpha$. For dataset with 1-digit weight = $\alpha$, we generate the dataset in a way such that for every mutliplicaiton, we first generate the number of digits for the two factors independently sampled from $\{1, ..., 10\}$ with weight $\{\alpha, 1, 1, ..., 1\}$.

phase of the $*$ operation. Remarkably, the model adeptly forges this connection, resulting in enhanced performance.

This shows that while including simple cases is crucial for the model to solve the hard problem, fostering connections between the simple case and the harder case can be an even more essential step. Exposure to the harder problem when presenting the simple case can be an efficient way to

| Data Format | Example |
|---|---|
| $n \times 1$ multiplication every 3 datapoints (with reversed product) | 6 5 1 2 5 * 6 # 0 5 7 0 9 3 |
| | 5 1 4 * 5 9 6 9 # 6 6 0 8 6 0 3 |
| | 3 8 6 3 1 * 6 1 6 # 6 9 6 6 9 7 3 2 |
| | 2 2 * 9 # 8 9 1 |
| | 7 9 8 0 4 * 8 8 8 9 2 # 8 6 1 7 3 9 3 9 0 7 |
| | 1 9 6 6 5 * 9 7 0 # 0 5 0 5 7 0 9 1 |
| | 2 6 4 9 * 6 # 4 9 8 5 1 |
| | 4 3 5 0 * 8 8 1 5 # 0 5 2 5 4 3 8 3 |
| | 2 0 2 0 * 9 9 8 9 # 0 8 7 7 7 1 0 2 |
| | 6 2 2 7 4 * 5 # 0 7 3 1 1 3 |
| | ... |
| First-step multiplication every 3 datapoints (with reversed product) | 6 5 1 2 5 * 1 5 3 0 6 % 0 5 7 0 9 3 |
| | 5 1 4 * 5 9 6 9 # 6 6 0 8 6 0 3 |
| | 3 8 6 3 1 * 6 1 6 # 6 9 6 6 9 7 3 2 |
| | 2 2 * 8 9 % 8 9 1 |
| | 7 9 8 0 4 * 8 8 8 9 2 # 8 6 1 7 3 9 3 9 0 7 |
| | 1 9 6 6 5 * 9 7 0 # 0 5 0 5 7 0 9 1 |
| | 2 6 4 9 * 6 7 9 6 % 4 9 8 5 1 |
| | 4 3 5 0 * 8 8 1 5 # 0 5 2 5 4 3 8 3 |
| | 2 0 2 0 * 9 9 8 9 # 0 8 7 7 7 1 0 2 |
| | 6 2 2 7 4 * 9 5 % 0 7 3 1 1 3 |
| | ... |

**Table 12:** Examples of datasets with $n \times 1$ multiplication every 3 datapoints and first-step multiplication every 3 datapoints.

associate the two. Our findings suggest that showing solutions to sub-components of a complex issue can be a more effective instructional approach than merely presenting elementary problems. Indeed, constructing datasets around these sub-problems might be more straightforward than building an array of incrementally challenging tasks, when aiming for a specific complexity level. For example, if we want to solve high-school math problems, we can easily collect a dataset consisting of high-school examination problems, but it can be hard to collect a series of problems with increasing difficulty that culminates at the high school level. Using primary school math as foundational problems might not guarantee a seamless transition in complexity or methodology. Instead, decomposing high school problems into their core sub-issues may offer a more coherent learning trajectory. As a result, subproblems of the hard problems can be an efficient simple case.

## B  ADDITIONAL DETAILS FOR SECTION 2

**Setup**    For all experiments in this section, our training data set consists of 300k samples, where the number of digits each factor has is sampled independently from a uniform distribution on $\{1, ..., n\}$. We call such a dataset a $n$-maximum-digit dataset. We train a random initialized GPT2-small for 300 epochs with a learning rate $2e - 5$. We test on 100 samples for each $(m_1, m_2)$ combination, where $m_1$ and $m_2$ are the number of digits in the two factors.

## C  ADDITIONAL DETAILS FOR SECTION 3

### C.1  ADDITIONAL DETAILS FOR SECTION 3.1

1. **Basic:** We begin by writing down the two addends. Next, for each digit $i$, we record the equation $c_i + a_i + b_i = s_i$, where $a_i$ represents the $i$-th digit of the first addend $a$, $b_i$ represents the $i$-th digit of the second addend $b$, $c_i$ represents the carry from the previous

1

**Table 13:** 10 digits maximum

| # Digits | 1 | 2 | 3 | 4 | 5 | 6 | 7 | 8 | 9 | 10 |
|---|---|---|---|---|---|---|---|---|---|---|
| 1 | 1.00 | 1.00 | 1.00 | 1.00 | 1.00 | 1.00 | 1.00 | 1.00 | 1.00 | 0.99 |
| 2 | 1.00 | 0.95 | 0.91 | 0.87 | 0.85 | 0.77 | 0.68 | 0.68 | 0.67 | 0.64 |
| 3 | 1.00 | 0.95 | 0.62 | 0.32 | 0.30 | 0.19 | 0.15 | 0.10 | 0.08 | 0.07 |
| 4 | 1.00 | 0.90 | 0.38 | 0.09 | 0.01 | 0.00 | 0.01 | 0.00 | 0.01 | 0.00 |
| 5 | 1.00 | 0.85 | 0.22 | 0.05 | 0.01 | 0.00 | 0.00 | 0.00 | 0.00 | 0.00 |
| 6 | 1.00 | 0.76 | 0.18 | 0.00 | 0.01 | 0.00 | 0.00 | 0.00 | 0.00 | 0.00 |
| 7 | 0.99 | 0.83 | 0.16 | 0.02 | 0.00 | 0.00 | 0.00 | 0.00 | 0.00 | 0.00 |
| 8 | 1.00 | 0.75 | 0.11 | 0.00 | 0.00 | 0.00 | 0.00 | 0.00 | 0.00 | 0.00 |
| 9 | 1.00 | 0.72 | 0.06 | 0.00 | 0.00 | 0.00 | 0.00 | 0.00 | 0.00 | 0.00 |
| 10 | 0.99 | 0.63 | 0.08 | 0.00 | 0.00 | 0.00 | 0.00 | 0.00 | 0.00 | 0.00 |

**Table 14:** Testing accuracy on models trained on 10-digit-maximum dataset in Basic format on 100 testing samples.

    digit, and $s_i$ represents the sum of $a_i$, $b_i$, and $c_i$. Finally, we write down the sum of the two numbers.

2. **Random Space:** We introduce random spaces between any two characters in the sentence in the form of "Basic". For each position, we generate a random space with a probability of 0.3.

3. **Recursive Scratchpad:** We modify the basic format so that before writing down the digit-wise sum $a_i + b_i + c_i = s_i$, we first record the remaining digits of $a$ and $b$ that haven't been included in their digit-wise sums, denoted as $a_n...a_i$ and $b_n...b_i$. Then, we write down the digit-wise sum for digit $i$, followed by the digit-wise sums obtained so far, $s_i...s_2 s_1$. In addition, we reverse the order of the digits in the two addends.

**Data Generation** For the training data, we first choose the number of digits for the two addends, denoted as $n$, uniformly random from $\{2, ..., 10\}$. Then, given $n$, we generate the two addends independently using a uniform distribution on $[0, 1, ..., 10^n - 1]$. Then, for each pair of addends, we generate the complete scratchpad steps according to the chosen data format. We generate a total of 120k samples independently following this process. For the testing data, for each number of digits $n$, we sample two addends uniformly random from $[10^{n-1}, ..., 10^n - 1]$ independently.

**Setup** We fine-tune pretrained GPT2-small(124M) for five epochs with learning rate $2e - 5$ on data in the three types of data formats in Table 7. The testing set consists of 100 $n$-digit plus $n$ digit additions with $n \in 9, 10, ...13$.

## C.2 Additional Details for Section 3.2

**Data Generation** For the training data, we first choose the number of digits, denoted as $n$, uniformly random from $\{2, ..., 10\}$. Then, given $n$, we generate number independently using from uniform distribution on $[0, 1, ..., 10^n - 1]$. We generate a total of 120k samples independently following this process.

We consider two types of testing data, to generate the regular testing data, for each digit $i$, we sample the number uniformly random from $[10^{n-1}, ..., 10^n - 1]$ independently. To generate the testing data with repetitive digits, for each digit $i$, we sample a digit from $[0, 1, ..., 9]$ and repeat the sampled digit $i$ times to form a number with $i$ digits.

## D Additional Details for Section 4

Formally, we have the following two types of data.

**Dialogue Data** We use GPT3.5 to generate the natural language data. We use the following prompt:

**(a)** Add Padding

| # Digits | 1 | 2 | 3 | 4 | 5 | 6 | 7 | 8 | 9 | 10 |
|---|---|---|---|---|---|---|---|---|---|---|
| 1 | 1.00 | 1.00 | 1.00 | 1.00 | 1.00 | 1.00 | 1.00 | 1.00 | 1.00 | 0.99 |
| 2 | 1.00 | 1.00 | 0.98 | 0.97 | 0.93 | 0.84 | 0.78 | 0.83 | 0.80 | 0.77 |
| 3 | 1.00 | 1.00 | 0.91 | 0.77 | 0.78 | 0.71 | 0.69 | 0.63 | 0.55 | 0.59 |
| 4 | 1.00 | 0.96 | 0.88 | 0.75 | 0.70 | 0.68 | 0.57 | 0.49 | 0.41 | 0.36 |
| 5 | 1.00 | 0.89 | 0.78 | 0.73 | 0.70 | 0.51 | 0.42 | 0.43 | 0.47 | 0.29 |
| 6 | 1.00 | 0.87 | 0.79 | 0.59 | 0.53 | 0.52 | 0.46 | 0.38 | 0.43 | 0.21 |
| 7 | 1.00 | 0.84 | 0.64 | 0.62 | 0.57 | 0.45 | 0.37 | 0.30 | 0.28 | 0.19 |
| 8 | 1.00 | 0.83 | 0.58 | 0.52 | 0.50 | 0.25 | 0.31 | 0.21 | 0.16 | 0.09 |
| 9 | 1.00 | 0.82 | 0.50 | 0.45 | 0.45 | 0.30 | 0.26 | 0.09 | 0.07 | 0.09 |
| 10 | 0.99 | 0.80 | 0.55 | 0.38 | 0.28 | 0.25 | 0.17 | 0.15 | 0.02 | 0.00 |

**(b)** Reverse Product

| # Digits | 1 | 2 | 3 | 4 | 5 | 6 | 7 | 8 | 9 | 10 |
|---|---|---|---|---|---|---|---|---|---|---|
| 1 | 1.00 | 1.00 | 1.00 | 1.00 | 1.00 | 1.00 | 1.00 | 1.00 | 1.00 | 1.00 |
| 2 | 1.00 | 1.00 | 0.99 | 0.89 | 0.27 | 0.14 | 0.11 | 0.11 | 0.10 | 0.15 |
| 3 | 1.00 | 1.00 | 0.92 | 0.26 | 0.08 | 0.01 | 0.02 | 0.02 | 0.00 | 0.01 |
| 4 | 1.00 | 0.99 | 0.24 | 0.04 | 0.00 | 0.00 | 0.00 | 0.00 | 0.00 | 0.00 |
| 5 | 1.00 | 0.68 | 0.10 | 0.00 | 0.01 | 0.00 | 0.00 | 0.00 | 0.00 | 0.00 |
| 6 | 1.00 | 0.45 | 0.05 | 0.01 | 0.01 | 0.00 | 0.00 | 0.00 | 0.00 | 0.00 |
| 7 | 1.00 | 0.29 | 0.04 | 0.00 | 0.00 | 0.00 | 0.00 | 0.00 | 0.00 | 0.00 |
| 8 | 1.00 | 0.22 | 0.04 | 0.00 | 0.00 | 0.00 | 0.00 | 0.00 | 0.00 | 0.00 |
| 9 | 1.00 | 0.16 | 0.01 | 0.00 | 0.00 | 0.00 | 0.00 | 0.00 | 0.00 | 0.00 |
| 10 | 1.00 | 0.13 | 0.01 | 0.00 | 0.00 | 0.00 | 0.00 | 0.00 | 0.00 | 0.00 |

**(c)** Reverse Product + Add Padding

| # Digits | 1 | 2 | 3 | 4 | 5 | 6 | 7 | 8 | 9 | 10 |
|---|---|---|---|---|---|---|---|---|---|---|
| 1 | 1.00 | 1.00 | 1.00 | 1.00 | 1.00 | 1.00 | 1.00 | 1.00 | 1.00 | 1.00 |
| 2 | 1.00 | 1.00 | 1.00 | 1.00 | 1.00 | 1.00 | 1.00 | 1.00 | 1.00 | 1.00 |
| 3 | 1.00 | 1.00 | 1.00 | 1.00 | 1.00 | 1.00 | 1.00 | 1.00 | 1.00 | 1.00 |
| 4 | 1.00 | 1.00 | 1.00 | 1.00 | 1.00 | 1.00 | 1.00 | 1.00 | 1.00 | 1.00 |
| 5 | 1.00 | 1.00 | 1.00 | 1.00 | 1.00 | 1.00 | 1.00 | 1.00 | 1.00 | 1.00 |
| 6 | 1.00 | 1.00 | 1.00 | 1.00 | 1.00 | 1.00 | 1.00 | 1.00 | 1.00 | 1.00 |
| 7 | 1.00 | 1.00 | 1.00 | 1.00 | 1.00 | 1.00 | 1.00 | 1.00 | 1.00 | 1.00 |
| 8 | 1.00 | 1.00 | 1.00 | 1.00 | 1.00 | 1.00 | 1.00 | 1.00 | 0.98 | 0.99 |
| 9 | 1.00 | 1.00 | 1.00 | 1.00 | 1.00 | 1.00 | 1.00 | 0.98 | 1.00 | 0.96 |
| 10 | 1.00 | 1.00 | 1.00 | 1.00 | 1.00 | 1.00 | 1.00 | 0.99 | 0.97 | 0.94 |

**Table 15:** Testing accuracy for models trained on data with padding and(or) reversed product when the maximum number of digits is 10.

*Create 20 dialogues between a student and a teacher where the student asks the teacher the sum of two numbers. Only use two numbers! An example of such dialogue is "Student: Hi, can you help me add two numbers, 34 and 432? Teacher: Sure. 466 is the answer." The teacher and the student are very talkative, so the dialogue contains longer sentences than my example. Please make sure your number is random and that different dialogues have different styles. The student asks the question in different ways in each dialogue and the teacher answers it in different ways. Only show the dialogues one by one. Do not show any numbering or anything else. Only use numbers in the list L.*

| # Digits | 1 | 2 | 3 | 4 | 5 | 6 | 7 | 8 | 9 | 10 | 11 | 12 | 13 | 14 | 15 | 16 | 17 | 18 | 19 | 20 |
|---|---|---|---|---|---|---|---|---|---|---|---|---|---|---|---|---|---|---|---|---|
| 1 | 1.00 | 1.00 | 1.00 | 1.00 | 1.00 | 1.00 | 1.00 | 1.00 | 1.00 | 1.00 | 1.00 | 1.00 | 1.00 | 1.00 | 1.00 | 1.00 | 1.00 | 1.00 | 1.00 | 1.00 |
| 2 | 1.00 | 1.00 | 1.00 | 1.00 | 1.00 | 1.00 | 1.00 | 1.00 | 1.00 | 1.00 | 1.00 | 1.00 | 1.00 | 1.00 | 1.00 | 1.00 | 1.00 | 1.00 | 1.00 | 0.46 |
| 3 | 1.00 | 1.00 | 1.00 | 1.00 | 1.00 | 1.00 | 1.00 | 1.00 | 0.99 | 1.00 | 1.00 | 1.00 | 0.99 | 1.00 | 1.00 | 1.00 | 1.00 | 1.00 | 0.72 | 0.03 |
| 4 | 1.00 | 1.00 | 1.00 | 0.81 | 0.12 | 0.03 | 0.00 | 0.00 | 0.00 | 0.00 | 0.00 | 0.00 | 0.00 | 0.00 | 0.00 | 0.00 | 0.00 | 0.00 | 0.00 | 0.00 |
| 5 | 1.00 | 1.00 | 1.00 | 0.05 | 0.02 | 0.00 | 0.00 | 0.00 | 0.00 | 0.00 | 0.00 | 0.00 | 0.00 | 0.00 | 0.00 | 0.00 | 0.00 | 0.00 | 0.00 | 0.00 |
| 6 | 1.00 | 1.00 | 1.00 | 0.00 | 0.00 | 0.00 | 0.00 | 0.00 | 0.00 | 0.00 | 0.00 | 0.00 | 0.00 | 0.00 | 0.00 | 0.00 | 0.00 | 0.00 | 0.00 | 0.00 |
| 7 | 1.00 | 1.00 | 0.99 | 0.00 | 0.00 | 0.00 | 0.00 | 0.00 | 0.00 | 0.00 | 0.00 | 0.00 | 0.00 | 0.00 | 0.00 | 0.00 | 0.00 | 0.00 | 0.00 | 0.00 |
| 8 | 1.00 | 1.00 | 1.00 | 0.00 | 0.00 | 0.00 | 0.00 | 0.00 | 0.00 | 0.00 | 0.00 | 0.00 | 0.00 | 0.00 | 0.00 | 0.00 | 0.00 | 0.00 | 0.00 | 0.00 |
| 9 | 1.00 | 1.00 | 1.00 | 0.00 | 0.00 | 0.00 | 0.00 | 0.00 | 0.00 | 0.00 | 0.00 | 0.00 | 0.00 | 0.00 | 0.00 | 0.00 | 0.00 | 0.00 | 0.00 | 0.00 |
| 10 | 1.00 | 1.00 | 1.00 | 0.00 | 0.00 | 0.00 | 0.00 | 0.00 | 0.00 | 0.00 | 0.00 | 0.00 | 0.00 | 0.00 | 0.00 | 0.00 | 0.00 | 0.00 | 0.00 | 0.00 |
| 11 | 1.00 | 1.00 | 1.00 | 0.00 | 0.00 | 0.00 | 0.00 | 0.00 | 0.00 | 0.00 | 0.00 | 0.00 | 0.00 | 0.00 | 0.00 | 0.00 | 0.00 | 0.00 | 0.00 | 0.00 |
| 12 | 1.00 | 1.00 | 1.00 | 0.00 | 0.00 | 0.00 | 0.00 | 0.00 | 0.00 | 0.00 | 0.00 | 0.00 | 0.00 | 0.00 | 0.00 | 0.00 | 0.00 | 0.00 | 0.00 | 0.00 |
| 13 | 1.00 | 1.00 | 0.98 | 0.00 | 0.00 | 0.00 | 0.00 | 0.00 | 0.00 | 0.00 | 0.00 | 0.00 | 0.00 | 0.00 | 0.00 | 0.00 | 0.00 | 0.00 | 0.00 | 0.00 |
| 14 | 1.00 | 1.00 | 1.00 | 0.00 | 0.00 | 0.00 | 0.00 | 0.00 | 0.00 | 0.00 | 0.00 | 0.00 | 0.00 | 0.00 | 0.00 | 0.00 | 0.00 | 0.00 | 0.00 | 0.00 |
| 15 | 1.00 | 1.00 | 1.00 | 0.00 | 0.00 | 0.00 | 0.00 | 0.00 | 0.00 | 0.00 | 0.00 | 0.00 | 0.00 | 0.00 | 0.00 | 0.00 | 0.00 | 0.00 | 0.00 | 0.00 |
| 16 | 1.00 | 1.00 | 1.00 | 0.00 | 0.00 | 0.00 | 0.00 | 0.00 | 0.00 | 0.00 | 0.00 | 0.00 | 0.00 | 0.00 | 0.00 | 0.00 | 0.00 | 0.00 | 0.00 | 0.00 |
| 17 | 1.00 | 1.00 | 1.00 | 0.00 | 0.00 | 0.00 | 0.00 | 0.00 | 0.00 | 0.00 | 0.00 | 0.00 | 0.00 | 0.00 | 0.00 | 0.00 | 0.00 | 0.00 | 0.00 | 0.00 |
| 18 | 1.00 | 1.00 | 1.00 | 0.00 | 0.00 | 0.00 | 0.00 | 0.00 | 0.00 | 0.00 | 0.00 | 0.00 | 0.00 | 0.00 | 0.00 | 0.00 | 0.00 | 0.00 | 0.00 | 0.00 |
| 19 | 1.00 | 1.00 | 0.74 | 0.00 | 0.00 | 0.00 | 0.00 | 0.00 | 0.00 | 0.00 | 0.00 | 0.00 | 0.00 | 0.00 | 0.00 | 0.00 | 0.00 | 0.00 | 0.00 | 0.00 |
| 20 | 1.00 | 0.61 | 0.05 | 0.00 | 0.00 | 0.00 | 0.00 | 0.00 | 0.00 | 0.00 | 0.00 | 0.00 | 0.00 | 0.00 | 0.00 | 0.00 | 0.00 | 0.00 | 0.00 | 0.00 |

**Table 16:** Testing accuracy for 20-maximum-digit with padding and reversed product.

| Data Format | Failure Cases |
|---|---|
| Basic (11 digits) | 9 0 8 9 4 0 3 0 2 8 7 + 6 7 9 2 4 1 6 0 2 8 1 : 
 0 + 7 + 1 = 8 , 0 + 2 + 8 = 1 0 , 1 + 0 + 0 = 1 , 0 + 3 + 6 = 9 , 0 + 0 + 1 = 1 , 
 0 + 4 + 4 = 8 , 0 + 9 + 2 = 1 1 , 1 + 8 + 9 = 1 8 , 1 + 0 + 7 = 8 , 0 + 9 + 6 = 1 5 , 
 1 + 0 + 0 = 1 , : 1 5 8 8 1 8 1 9 1 0 8 |
| Basic (11 digits) | 9 7 4 0 4 8 4 9 0 7 0 + 6 3 2 4 3 4 8 2 9 9 7 : 
 0 + 0 + 7 = 7 , 0 + 0 + 9 = 9 , 0 + 9 + 2 = 1 1 , 1 + 4 + 8 = 1 3 , 1 + 8 + 4 = 1 3 , 
 1 + 4 + 3 = 8 , 0 + 0 + 4 = 4 , 0 + 4 + 2 = 6 , 0 + 7 + 3 = 1 0 , 1 + 9 + 6 = 1 6 , 
 1 + 0 + 0 = 1 , : 1 6 0 6 4 8 3 3 1 9 7 |
| Random Space 

 (12 digits) | 2   3 6   6 7 6     8 3 6 0 8   9   + 1   7 0 7 1 2 0   5 9 6   3 0 : 0 + 9   + 0 = 9 , 
  0 +      ... (Empty spaces) |
| Recursive Scratchpad 
 (13 digits) | 7 3 7 8 0 5 5 5 9 0 1 8 9 + 8 7 8 1 7 8 2 4 6 5 9 7 9 : 0 + 7 + 8 = 1 5 , = 5 , 
 3 7 8 0 5 5 5 9 0 1 8 9 + 7 8 1 7 8 2 4 6 5 9 7 9 : 1 + 3 + 7 = 1 1 , = 1 5 , 
 7 8 0 5 5 5 9 0 1 8 9 + 8 1 7 8 2 4 6 5 9 7 9 : 1 + 7 + 8 = 1 6 , = 6 1 5 , 
 8 0 5 5 5 9 0 1 8 9 + 1 7 8 2 4 6 5 9 7 9 : 1 + 8 + 1 = 1 0 , = 0 6 1 5 , 
 0 5 5 5 9 0 1 8 9 + 7 8 2 4 6 5 9 7 9 : 1 + 0 + 7 = 8 , = 8 0 6 1 5 , 
 5 5 5 9 0 1 8 9 + 8 2 4 6 5 9 7 9 : 0 + 5 + 8 = 1 3 , = 3 8 0 6 1 5 , 
 5 5 9 0 1 8 9 + 2 4 6 5 9 7 9 : 1 + 5 + 2 = 8 , = 8 3 8 0 6 1 5 , 
 5 9 0 1 8 9 + 4 6 5 9 7 9 : 0 + 5 + 4 = 9 , = 9 8 3 8 0 6 1 5 , 
 9 0 1 8 9 + 6 5 9 7 9 : 0 + 9 + 6 = 1 5 , = 5 9 8 3 8 0 6 1 5 , 
 0 1 8 9 + 5 9 7 9 : 1 + 0 + 5 = 6 , = 6 5 9 8 3 8 0 6 1 5 , 
 1 8 9 + 9 7 9 : 0 + 1 + 9 = 1 0 , = 0 6 5 9 8 3 8 0 6 1 5 , 
 8 9 + 7 9 : 1 + 8 + 7 = 1 6 , = 6 0 6 5 9 8 3 8 0 6 1 5 , 
 9 + 9 : 1 + 9 + 9 = 1 9 , = 9 6 5 9 8 3 8 0 6 1 5 , = 1 9 6 5 9 8 3 8 0 6 1 5 |

**Table 17:** Sample failure cases of the three data formats.

We replace $L$ with 50 random numbers. We generate $L$ in the following way. For 2-to-3-digit addition, we generate each element of $L$ i.i.d 2-digit numbers uniformly with probability 0.3, and i.i.d. 3-digit numbers uniformly with probability 0.7. For 4-to-5-digit addition, we generate each element of $L$ i.i.d 4-digit numbers uniformly with probability 0.5, and i.i.d. 4-digit numbers uniformly with probability 0.5. We generate around 12200 dialogues on 2-to-3-digit addition and 1040 dialogues on 4-to-5-digit addition. In this way, our natural language data contains many more 2-to-3-digit numbers than 4-to-5-digit numbers. Our training data contains only one dialogue in one sentence. Although we specify in the prompt the format, around 3% of the data does not follow the prompt and output the dialogue in two lines or output a separator between two dialogues. For the 4-to-5-digit addition, there is an error rate of $0.2\%$. We did not correct these errors to reflect the noisy data collected in practice.

**Addition Data**    We study two types of arithmetic data.

1. **Basic** For two numbers $a$ and $b$ with $n$ and $m$ digits, $a_n...a_1$ and $b_m...b_1$. We write down $a_n...a_1 + b_m...b_1 = s_l s_{l-1}...s_1$, where $s_l...s_1$ is the sum of $a$ and $b$.

2. **Random Space** For two numbers $a$ and $b$, we first write down $a_n...a_1$, followed by $n_s$ many random spaces. Then, we write down $b_m...b_1$, followed $n_s'$ many random spaces. Finally, we write down the sum $s_l...s_1$. $n_s$ and $n_s'$ are uniform numbers from $\{1, ..., 5\}$.

For each type, generate 120k samples.

**Setup**    We compare the performance of the models trained on purely dialogue data and those trained the mixture of dialogue and addition data. To test the model in the dialogue context, we prompt the model using a simple student question, "Student: Hi, what is the sum of #a and #b?", where we replace $\#a$ and $\#b$ with the randomly generated numbers. To test the model in the pure arithmetic context we prompt the model using "#a + #b" or "#a   #b" depending on the data format. We check the accuracy of the teacher's response on 100 samples for each number of digits. We compare GPT2-small models with positional embedding, without positional embedding and with random embedding (see Section 3.2 for definition). The models with random embedding do not have positional embedding. For all the runs, we train from a randomly initialized model for 100 epochs with a learning rate 2e-5.

