# OpenReview forum: "Positional Description Matters for Transformers Arithmetic"
_ICLR.cc/2024/Conference — Submitted to ICLR 2024_

### Official Review · Reviewer_ywtf · 2023-10-26

**Soundness:** 3 good
**Presentation:** 2 fair
**Contribution:** 2 fair
**Rating:** 5
**Confidence:** 4

**Summary:**

The paper focuses on the arithmetics capabilities of autoregressive Transformers. Their hypothesis is that many of the failure modes of the networks are linked to the positional encodings. They propose various ways to break the over-reliance on absolute positional encodings, and they show that it improves performance. The authors also demonstrate that this, combined with training on both natural language and synthetic samples, improves the arithmetic capabilities of the models even when the task is formulated in natural language.

**Strengths:**

- The paper analyses under which conditions Transformers can learn addition and multiplication.
- The authors demonstrate multiple experiments showing that randomizing positions in various ways helps.
- The authors present a simple randomized positional embedding method.
- The authors show that mixed data training improves the performance of the arithmetics presented in natural language, but only if either no absolute position is used, or the positions are randomized.

**Weaknesses:**

# Clarity

The clarity of the writing can be improved. Many of the tables are missing information on what is the maximum number of steps the model was trained for (eg. Fig 1 and Fig 2). What is the "pretrained model" pre-trained on? How is it fine-tuned on the new data? The contribution of the paper could be clarified further. The conclusion section is missing (it is part of the previous section).

I think it would be worth discussing a broader view of these problems. E.g not all problems described here are directly related to positional encodings. For example, reversing the results has to do with the decomposition of the task, and the ability to use subresults of less significant digits to compute the more significant ones.

# Soundness

I'm not convinced by the connection between the recursive scratchpad. It could also be a completely orthogonal effect.

# Baselines

The authors only compare to absolute positional embedding baseline, which is known for weak generalization properties [1,2]. RoPE also tends to be not great in length extrapolation, but Transformer XL-style relative PE [3] tends to work well. The authors should consider comparing to these models.

# Novelty

It is unclear how to position this paper fits in the existing work considering algorithmic tasks and positional encodings. [3] show the benefits of randomized positional encodings on various algorithmic tasks and [4] shows the benefits of randomly shifted positional encodings. [5] investigates different number representations for arithmetic. [6] shows that with closeness bias instead of positional encodings, length extrapolation is possible on simple algorithmic tasks, like Listops. The advantages of using scratchpad and randomized positional encodings are known. It is also unclear how the findings of this paper transfer to relative positional encodings, which are used in most modern transformers.


1. Nogueira et al, 2023, The Impact of Positional Encoding on Length Generalization in Transformers
2. Csordas et al, 2021: The Devil is in the Detail: Simple Tricks Improve Systematic Generalization of Transformers
3.  Dai et al, 2019: Transformer-XL: Attentive Language Models Beyond a Fixed-Length Context
3. Ruoss et al, 2023: Randomized Positional Encodings Boost Length Generalization of Transformers
4. Kiyono et al, 2021: SHAPE: Shifted Absolute Position Embedding for Transformers
5. Nogueira: Investigating the Limitations of Transformers with Simple Arithmetic Tasks
6. Csordas et al, 2021: The Neural Data Router: Adaptive Control Flow in Transformers Improves Systematic Generalization

**Questions:**

- "In the “Basic” data format, the model fails to generalize numbers exceeding 10 digits."  - what was the length seen during the training?
- "training solely on arithmetic data can’t guarantee an accurate solution to the difficult problems due to the lack of difficult samples in the training set and the errors present" - aren't the arithmetic data more complicated, in the sense of more digits? Also, isn't it perfect, as it is synthetic?
- For the 15-digit multiplication not extrapolating to 20, I noticed that the performance breaks down much earlier than reaching even 15 digits. Isn't the problem that now the result is padded to 20 digits? What about training until 15 with a padding of 20, and then checking if it generalizes to 20.
- In Table 6, is there any performance difference in the "Basic" and "Random Space" versions between the pre-trained vs randomly initialized settings?

---

> ### Author Response · Authors · 2023-11-22
>
> We thank the reviewer for the helpful comments. The training setup for Fig 1 and 2 can be found in their captions. Pretrained means the default pretrained GPT2. All the other information on how it is fine-tuned can be found in the appendix.
>
> We believe reversing the result also falls under the main theme of positional information because reversing emphasizes the importance of putting the digits of different significance in the right positions.
>
> We appreciate the related works and the summary the reviewers mentioned. We would like to point out that while we proposed random embedding, we view it as a simple trick to solve the issues we noticed rather than an alternative to positional encoding. For this reason, we did not provide a comprehensive comparison between it and other existing encodings. However, we agree that a more comprehensive comparison is an important next step if we want to apply random embedding more widely.
>
> Response to the questions:
> - Up to 10 digits
> - This is a typo. It should be “the natural language data”. Thank you for pointing this out.
> - We didn’t try this, but we agree this could be an interesting experiment to do. Thank you for suggesting this.
> - We did not see any significant difference.

---

> > ### Comment · Reviewer_ywtf · 2023-11-22
> >
> > I would like to thank the authors for their response.
> >
> > > We thank the reviewer for the helpful comments. The training setup for Fig 1 and 2 can be found in their captions. Pretrained means the default pretrained GPT2. All the other information on how it is fine-tuned can be found in the appendix.
> >
> > Sorry, I used the wrong wording. It is not clear what the was maximum length seen during the training, which is a crucial point for understanding the plots.
> >
> > > We believe reversing the result also falls under the main theme of positional information because reversing emphasizes the importance of putting the digits of different significance in the right positions.
> >
> > I think this is more about the order of computation as opposed to positional encodings.
> >
> > > We would like to point out that while we proposed random embedding, we view it as a simple trick to solve the issues we noticed rather than an alternative to positional encoding.
> >
> > We can use the same argument for reversing the sequence order or padding. Actually, randomized positional encodings should be more universal, if they work (which is unclear at this point). Also, relative positional encodings are more universal, and their performance is unclear on the tasks the authors considered.

---

> > > ### Author Response · Authors · 2023-11-22
> > >
> > > Thank you for the quick response. The maximum length seen during training is 10 digits. We believe what we study is more general than positional encoding, so we use the word positional information. We agree that randomized positional encodings should be more universal and it's worthwhile to study its performance more comprehensively.

---

### Official Review · Reviewer_Nh7J · 2023-10-27

**Soundness:** 2 fair
**Presentation:** 2 fair
**Contribution:** 2 fair
**Rating:** 3
**Confidence:** 4

**Summary:**

In this paper, the authors study the performance of Transformer architectures on a variety of arithmetic problems. In a first set of simulations, they show that representing multi-digit multiplications using a tabular (padded) input format allows Transformers to extract the underlying multiplication procedure more easily, achieving high accuracy even with longer operands than those seen during training. In a second set of simulations, they investigate addition problems, and show that in this case it might be instead useful to add random spaces between the numbers to force the Transformer to learn the underlying procedure, rather than simply memorize the training patterns. They also propose to exploit random embeddings to improve generalization capabilities. In a final set of simulations, they nevertheless show that Transformers still struggle with math word problems (i.e., addition problems formulated by mixing natural and mathematical language).

**Strengths:**

-	The article is clear and well-written. The research questions are well-motivated.
- The authors carried out a substantial amount of work and present results from a variety of interesting analyses.
-	Focusing on “basic” architectures (GPT-2) and simple symbolic domains, such as arithmetic, allows to get useful insights about the computational capabilities of Transformers, which might then be extended to more complex architectures and reasoning tasks.

**Weaknesses:**

-	Related works could be expanded (see below). Furthermore, the training/testing setups used in the present work differ from those used in other similar work, making it more challenging to compare the current results with previous contributions.
-	Different Sections investigate different problems (e.g., multiplication vs. addition vs. math word problems), often by introducing opposite approaches (e.g., padding vs. insertion of random spaces) or by relying on quite different training regimens (e.g., train on 5-digit and test on 15-digits multiplication problems vs. train on 10-digit and test on 12-digit addition problems). This makes it difficult to frame the contribution of the paper, which seems a collection of heterogeneous experiments rather than a coherent proposal for a unified framework.
-	Padding the input is a simple method, which allows to reach an impressive boost in 15-digits multiplication. However, it requires specifying the maximum length of the operands (and if this gets overestimated, padding becomes quite inefficient). Furthermore, in a natural language setting (e.g., Section 4) the Transformer should learn when and how to pad, which seems quite implausible as a real use case.
-	The “generalization” capabilities described in Section 3 (from 10-digits to 11- or 12-digits) do not seem to indicate the emergence of a systematic addition algorithm.
-	The paper does not include any Reproducibility Statement or any pointer to source code repositories, which makes it difficult to replicate the simulations and the experimental setup.

**Questions:**

-	Related works: The authors argue that no previous work has demonstrated that Transformers can compute the product of two 15-digit number multiplication. A recent paper (already mentioned by the authors: https://arxiv.org/abs/2306.15400) has shown that by adding a tiny number of long sequences to the training set allows generalization to 35-digit operands in multiplication tasks (though it should be stressed that in this case we are injecting some OOD cases into the training set). Another work that might be worth mentioning has shown that the “surface representation” of the input has a significant impact on the Transformer generalization capability in arithmetic tasks (https://arxiv.org/abs/2102.13019), which is partially aligned with some findings observed in the present paper. Since the number of papers focusing on arithmetic tasks has significantly increased in the past few years, the authors might also consider mentioning recent reviews on the topic (e.g., https://arxiv.org/abs/2303.07735).
-	The generalization capabilities on addition problems should be tested on a wider set of cases, involving much longer operands and possibly also a larger number of operands (see, for example, https://arxiv.org/abs/2207.02536). High accuracy in the out-of-distribution ranges would more convincingly show that the data format / positional embedding can be successfully used to systematically grasp the problem structure.
-	The authors seem to implement a sort of curriculum learning strategy in their approach, since learning starts from “easy” problems (such as one-digit multiplication) and progressively covers more complex problems. However, such training strategy has not been systematically described, nor compared with an alternative strategy not based on curriculum learning.
-	It would be useful to compare the proposed “random embeddings” with other recent proposals that have been shown to improve numerical reasoning tasks, such as label positional encodings (https://arxiv.org/abs/2210.00400).
-	English phrasing and grammar are fine, but there are a few typos and a few sentences that can be improved (e.g., “To solve this issue, we propose a simple solution targeting this issue”).

---

> ### Author Response · Authors · 2023-11-22
>
> We thank the reviewer for the helpful comments. We appreciate the related works the reviewer mentioned and will add them to our revision. We agree with some of the limitations the reviewer mentioned in the weakness section, e.g., the lack of a unified framework for different problems and the requirement to specify the maximum length of padding. However, we believe the approaches studied in this paper, though have various limitations, could inspire future work that overcomes the limitations. Although the generalization capabilities are only 11- or 12-digit, it indicates a phase transition from no generalization at all to some degree of generalization, which is already challenging as shown in previous works[Lee et al., 2023].
>
> We thank the reviewer for pointing out https://arxiv.org/abs/2207.02536 and suggesting potential other cases that could test generalization capabilities.
>
> The multiplication studied in https://arxiv.org/abs/2306.15400 is significantly simpler than the multiplication studied in our paper. In their work, one of the factors is always smaller than 1000. However, the complexity of large-number multiplication is mainly due to both factors being large.
>
> We did not use any curriculum learning strategy. The model is trained on all samples in the dataset in each epoch. We used the concepts of easy and hard problems for an easier understanding of the experiment results.

---

### Official Review · Reviewer_pA8q · 2023-11-01

**Soundness:** 3 good
**Presentation:** 2 fair
**Contribution:** 2 fair
**Rating:** 3
**Confidence:** 4

**Summary:**

Language models struggle to solve arithmetic tasks, especially as the number of digits increases. This paper looks at the effect of data format and positional encoding on this struggle. Particularly, when doing multiplication, it is shown that if the numbers are padded to the inference length the model can perform multiplications of numbers that have many more digits than those observed in training (a reversing trick is also applied so that the model outputs the digits in order of calculation, i.e. the rightmost digit first followed by the tenth and hundredth and so on). While padding removes the need for extrapolation, adding random spaces is proposed as a way to make the model more resilient and generalizing. Finally, without the aforementioned data format, it is shown that using alternatives of the absolute positional encoding can allow the model to generalize to lengths not observed during training. Indeed the authors show successful extrapolation with a pre-trained model without positional embedding (though this fails if it is trained from scratch). An alternative positional encoding (random encoding) is also proposed which has superior extrapolation results on simple tasks such as addition and reversing numbers.

**Strengths:**

The paper cleverly uses simple settings that isolate the problem and allow testing the effect of different solutions. The results, such as the ability to extrapolate simply through padding, are also quite interesting and simple enough that can be implemented (even if only as a temporary solution) to allow models perform well on a larger set of numbers.

The investigation into the effect of positional encoding also takes the effect of pre-training into account and points out that using no positional encoding works only when the model is pre-trained.

The paper and description of the experiments are in most parts clear and well justified.

**Weaknesses:**

1. The paper is not structured very cohesively. There is also no conclusion or future work section which makes the paper a bit incomplete. The proposed solutions are more experimental and it is not clear which ones can be readily applied in practice.

2. My understanding is that the problem with extrapolation seems to lie with absolute positional encoding. However, there are many other encodings already designed and being used in practice including relative ones such as rotary. It is possible that the issue is actually the learnable positional encoding in which case the sinusoidal encoding proposed in the original Transformers paper is available. In light of these, it is not clear why yet another encoding (random encoding) is proposed.

3. Following up on the above note, when proposing a new encoding, it should be compared with other existing encodings. Furthermore such comparison should not be limited to arithmetic tasks but should also include other tasks (e.g. normal next token predictions). While the paper includes experiments for length extrapolation with dialogue data, the side-effect of training without positional encoding or with the random encoding on other tasks (non-arithmetic) is not considered.


Some minor comments:
1. The reverse task with repeats is not explained anywhere.
2. In Table 7 it seems there is a mismatch in the intermediary steps. In particular it is written 32 + 28 while the initial addition is 239 + 821. if I understand correctly either both should be reversed or neither?

**Questions:**

1. Recently [1] argued that using no positional encoding works well on various language modeling tasks. Since your results suggest that it doesn't work for arithmetic tasks, how does your results compare?

2. The idea of adding spaces to make the model resilient is interesting. However, how does this compare with recently proposed methods to allow extrapolation such as Ruoss. et. al's work of using random positions?

3. When you add multiple spaces between two digits, does it necessarily mean more tokens?

---

> ### Author Response · Authors · 2023-11-22
>
> We thank the reviewer for the helpful comments. While we did not write a separate conclusion section, we summarized our work and pointed out future directions in the introduction. We proposed random encoding as a simple trick for the problem studied in this paper, rather than a new encoding that would work for all tasks. For this reason, we did not compare it with other existing encodings on other tasks. However, we agree it would be worthwhile to test the random embedding on other tasks as the next step.
>
> Response to the minor comments.
>
> 1. Section C.2 setup (the last sentence) describes the reverse task with repeats.
> 2. In Table 7, we reverse the digits in all summands except the two summands at the beginning of the sentence.
>
> Response to the questions.
> 1. The reference [1] is missing.
> 2. Adding random space modifies the data representation, which is a fundamentally different approach and can be much easier than modifying the positional encoding.
> 3. Yes. It would be interesting to see if this can be done more efficiently.

---

### Official Review · Reviewer_34Rb · 2023-11-06

**Soundness:** 3 good
**Presentation:** 2 fair
**Contribution:** 2 fair
**Rating:** 5
**Confidence:** 2

**Summary:**

This paper investigates a transformer's capabilities in multiplication of decimal integer numbers up to 15 digits in length (with some tests in higher ranges as well).

The authors create a large data set for such multiplication operations that is balanced with respect to the number of digits involved.

They show that when training transformers to perform multiplication it is beneficial to tokenize each digit (they enforce this not by specifying a token for each digit in the tokenizer but by adding spaces between numbers), feeding the output in opposite ordering (small decimal places first), aligning the times operator between examples using zero-padding. Incorporating all of these tricks they can train and test on up to 15 digit number multiplication and provide evidence of the performance in extensive tables.

Next they investigate length generalization. They find that standard positional embedding may be lacking, and remove it. In some settings random tags are added to the tokens in order to help the model be able to identify them within one sample. Both from-scratch training and fine-tuning are analyzed.

Further, they investigate the performance of transformers in the full-text context, and whether performance of a text transformer on multiplication can be improved by adding examples to the data set. They provide extensive experimental results.

**Strengths:**

This paper does extensive experimental work to investigate the described settings.

**Weaknesses:**

This paper contains flippant sentences such as
- "Our findings reveal that even such modest-sized models can adeptly execute intricate arithmetic tasks" (when they use GPT models of >100M parameters to solve a very straight-forward task. For comparison, for the case of addition, people have been able to train 1-layer transformers to do it)
- "In Section 2.1, we show a simple 12-layer transformer can output the product of 15 × 15-multiplication directly, demonstrating the immense potential of transformers." (the authors should provide a non-arbitrary definition of the word "simple" that safely contains a 12-layer transformer and not all other conceivable neural networks. Also is it then correct to assume 6-layer transformers cannot do this task?)
- "Large number multiplication is complicated, [...]" (it is not - it can be implemented as a "simple" convolution over the digits)
In general, it would be good if the authors provided a bit more grounding for their fact claims and value judgements. The reader's comprehension suffers from the lack of it. Thankfully these aspects are fixable by some language revision. The paper would in general benefit from removing unnecessary sentences and claims.

When positional encoding is removed and replaced by temporary random tags, it should be stated that no positional information is left other than the causal decoder mask. If the multiplication still works when removing positional embedding and the model is capable of not confusing the digits, the causal mask is the only possible origin of precise positional information. Please confirm if this assessment is correct. As a consequence, especially because the multiplication seemed to have run better in this case, please check if the multiplication mechanism is implemented such that the only meaningful operations happen at the border of the causal mask. This would also explain generalization, because it would lead to a digit position by digit position algorithm. This question should be answerable by studying attention head activations.

In general, while this paper does a large number of experiments, the reader is left very much in the dark about any mechanisms that might be involved here. Studying the mechanisms and not simply listing performance values would greatly enhance this contribution. As it stands, it feels like an arbitrary collection of experiments.

**Questions:**

Please see the questions listed in the weaknesses section.

---

> ### Author Response · Authors · 2023-11-22
>
> We thank the reviewer for the review and the helpful suggestions on writing. In the first sentence, “Intricate arithmetic tasks” is not limited to the addition task the reviewer points out. It includes all the tasks studied in the paper, especially large number multiplication. We are not aware of any previous work that is able to perform $15\times 15$-digit multiplication. For this reason, we also believe it is unfair for the reviewer to say large number multiplication is a simple task when pointing out the third sentence. In the second sentence, “simple” means the model uses the original architecture of the transformers with no additional components. It does not imply 6-layer transformers cannot do this task. We can avoid this word in our revision.
>
> We agree the causal mask is the origin of positional information when the absolute positional embedding is removed. However, we did not have any results showing multiplication is better in this case.
>
> Throughout the paper, we have discussions on the mechanisms involved in the experiment result we show in this paper. We believe it would be helpful if the reviewer could point out more concretely which mechanism could be discussed in more detail.

---

### Meta-Review · Area_Chair_NFKX · 2023-12-01

**Metareview:**

While the paper considers an interesting problem, reviewers found the paper containing preliminary results and raised significant concerns over the overall story,  missing comparisons and limited discussion of related works. I agree with this assessment and find the paper to be not ready for publication at this point and recommend rejection.

**Justification For Why Not Higher Score:**

Preliminary results and missing comparisons.

**Justification For Why Not Lower Score:**

N/A

---

### Decision · Program_Chairs · 2024-01-16

Reject